# Application of Physiologically Based Pharmacokinetic Modeling to Predict Maternal Pharmacokinetics and Fetal Exposure to Oxcarbazepine

**DOI:** 10.3390/pharmaceutics14112367

**Published:** 2022-11-03

**Authors:** Lixia He, Meng Ke, Wanhong Wu, Jiarui Chen, Guimu Guo, Rongfang Lin, Pinfang Huang, Cuihong Lin

**Affiliations:** Department of Pharmacy, The First Affiliated Hospital of Fujian Medical University, 20 Cha Zhong M. Rd, Fuzhou 350005, China

**Keywords:** oxcarbazepine, pregnancy, pharmacokinetics, maternal–fetal, physiologically based pharmacokinetic model

## Abstract

Pregnancy is associated with physiological changes that may affect drug pharmacokinetics (PKs). The aim of this study was to establish a maternal–fetal physiologically based pharmacokinetic (PBPK) model of oxcarbazepine (OXC) and its active metabolite, 10,11-dihydro-10-hydroxy-carbazepine (MHD), to (1) assess differences in pregnancy, (2) predict changes in PK target parameters of these molecules following the current dosing regimen, (3) assess predicted concentrations of these molecules in the umbilical vein at delivery, and (4) compare different methods for estimating drug placental penetration. Predictions using the pregnancy PBPK model of OXC resulted in maternal concentrations within a 2-fold error, and extrapolation of the model to early-stage pregnancies indicated that changes in median PK parameters remained above target thresholds, requiring increased frequency of monitoring. The dosing simulation results suggested dose adjustment in the last two trimesters. We generally recommend that women administer ≥ 1.5× their baseline dose of OXC during their second and third trimesters. Test methods for predicting placental transfer showed varying performance, with the in vitro method showing the highest predictive accuracy. Exposure to MHD in maternal and fetal venous blood was similar. Overall, the above-mentioned models can enhance understanding of the maternal–fetal PK behavior of drugs, ultimately informing drug-treatment decisions for pregnant women and their fetuses.

## 1. Introduction

Oxcarbazepine (OXC) is approved for use alone or in combination with other antiepileptic drugs (AEDs) for primary generalized tonic double and partial seizures with or without secondary generalized seizures. Its chemical structure is similar to that of carbamazepine, but its metabolism is different. Carbamazepine is mainly metabolized by liver CYP3A4 to the active metabolite carbamazepine-10,11-epoxide, which is then excreted by the kidney through hydration [1]. OXC is rapidly absorbed and metabolized as its main clinical active metabolite, 10,11-dihydro-10-hydroxycarbazepine (monohydroxy derivative, MHD), through aldehyde ketone reductase (AKR1C1-4). Oral OXC formulations have high bioavailability (>95%). Peak concentration is obtained within 1–3 h after a single dose, whereas MHD peak concentration occurs within 4–12 h. At steady state, MHD peaks 2–4 h after drug administration. Approximately 40% of the MHD is bound to plasma proteins [2,3].

Clinically significant changes in the pharmacokinetics (PKs) of AEDs may complicate antiepileptic treatment in pregnant women [4,5]. Lower ratios to target concentration are associated with the worsening of seizures. Early therapeutic drug monitoring (TDM) and dose adjustment may help avoid increased seizure frequencies [6]. According to a review and analysis on the monitoring of therapeutic drugs before and after pregnancy in women with epilepsy by Arfman et al. [7], the trough concentration of AEDs should be monitored twice before pregnancy as a reference concentration. In the presence of risk factors for convulsions, dose adjustments during pregnancy based on TDM should be considered if AED concentrations are reduced by 15–25% from preconception reference concentrations, and dose adjustment is recommended if the change exceeds 25%. A recent retrospective analysis found significant changes in the PKs of multiple newer AEDs in women with epilepsy, underscoring the need for TDM during pregnancy [8]. TDM is encouraged at least once every three months prior to pregnancy, especially in women with poorly controlled seizures, and AED doses should be titrated to compensate for any changes in drug clearance [9,10].

Adequate control of seizures in pregnant women with epilepsy is critical because frequent and prolonged maternal seizures can have deleterious consequences, including miscarriage, fetal intracranial hemorrhage, and preterm birth. However, the teratogenicity of some older generation AEDs (such as phenytoin sodium, valproate sodium, etc.) and the potential induced pharmacokinetic changes in AEDs make the treatment of epilepsy during pregnancy complex [10]. The ideal effect of AEDs is to stably control epileptic seizures in the mother without adverse effects on the fetus. However, prenatal exposure to AEDs has been associated with a greater risk of major congenital malformations [11,12]. The risk of major congenital malformations is influenced not only by the type of AED but also by other variables, such as the dose. For example, a significant risk of malformation was observed with valproic acid and phenobarbital at all doses studied, with carbamazepine at doses greater than 400 mg/d, and with lamotrigine monotherapy at doses less than 300 mg/d [13]. In addition, exposure to antiepileptic drugs during pregnancy and lactation may also affect the development of children’s nervous systems. It is suggested that pregnant women should consider counseling to adjust to the safest treatment scheme [14].

The factors influencing pregnancy-induced pharmacokinetic changes when using AEDs include protein-binding rate, volume, and blood flow. A growing number of researchers have speculated that the decrease in plasma concentrations of the active metabolite of OXC during pregnancy may be due to changes in glucuronyltransferase activity [3,4,5,6,7,8,9,10]. Both OXC and MHD have significant placental transport [1], which may lead to excessive fetal plasma concentrations and result in serious adverse effects on the fetus. Limited data suggest that the administration of OXC during pregnancy can cause serious birth defects. The most common congenital malformations related to maternal OXC treatment include ventricular septal defect, cleft palate with cleft lip, and Down syndrome [10]. According to the North American AED Pregnancy Registry, women exposed to OXC during pregnancy had a relative risk of 1.6 (90% confidence interval (CI): 0.46–5.7) [15] for their child having congenital malformations compared with women not exposed to any AEDs during pregnancy. However, due to the small number of the study population, the teratogenic mechanism of OXC remains unclear, with fetal toxicity studies lacking. Therefore, a PK study of OXC in both the mother and fetus during pregnancy would have great reference value for fetal toxicity guidance and OXC dose adjustment.

Physiologically based pharmacokinetic (PBPK) modeling is a promising approach to overcome the limitations of traditional studies in analyzing the PKs of xenobiotics in special populations where clinical trials are difficult to conduct, such as pregnant women and their fetuses [16]. The purpose of our study was to characterize the maternal–fetal PKs of OXC during pregnancy and to provide recommendations for OXC dose adjustment.

## 2. Materials and Methods

### 2.1. Software, Clinical Data, and General Workflow

Open Systems Pharmacology Suite version 11.0 suite, including PK-Sim^®^ and MoBi^®^ (https://github.com/Open-Systems-Pharmacology, accessed on 25 May 2022), was used to perform the modeling work in pregnant women. The reported plasma time concentration data were digitized using GetData Graph Digitizer 2.26. Plots were created, and statistical analysis was performed using OriginPro^®^ (OriginLab, Northampton, MA, USA).

Detailed MHD clinical and TDM data in the First Affiliated Hospital of Fujian Medical University from January 2015 to May 2022 were obtained. The inclusion criteria were: (1) age ≥16 years, (2) epilepsy diagnosis that met the criteria published by the International League Against Epilepsy2014, (3) availability of at least one blood MHD concentration and corresponding oral daily dose,and (4) follow-up of every 3 months. After screening, a total of 121 patients met the requirements, including 54 women of gestational age.

Appendix A depicts the 27-compartment pregnancy model structure and general modeling workflow. As a first step, we constructed an adult PBPK model of OXC using the default 18-compartment model structure designed for small molecules [17].The model was validated using PK data in multiple settings, including hospital TDM data. Clinical PK data for model development and validation were identified and divided into test and validation sets (Table 1). The pregnancy intervals of nine pregnancy types and pregnancy-related anatomical/physiological changes were added through MoBi^®^. After the parameters, such as pregnancy-related protein binding, were modified, the model was transferred back to PK-Sim^®^ to be included in the pregnant population and modified into the corresponding gestational age parameters. The obtained PK data were used for verification, and the non-pregnant adult PBPK model was extrapolated to the pregnant population [8,9,10,11,12,13,14,15,16,17,18]. In vitro human placental perfusion data were incorporated into the pPBPK (pregnancy PBPK) model to estimate fetal PK. Placenta, fetal blood, tissues, and amniotic fluid chambers were added as models. The parameters of placental metastasis were estimated by different subarea models of perfusion data.

### 2.2. Development of a Non-Pregnant Population Model

The population PBPK model of non-women incorporated the PK-Sim^®^ standard model structure containing 18 organs and tissues as described previously [25,26,27,28,29]. The physical, chemical, and biochemical parameters of the OXC model are shown in Table 2. First, the basic parameters of OXC, such as its molecular weight, lipophilicity, protein-binding rate, and solubility, were input into the model. The Weibull 50% dissolution time (in minutes) was generated by internal optimization of the software, and the tissue plasma partition coefficient was calculated using the Berezhkovskiy method [30,31]. OXC is rapidly and almost completely converted into a pharmacologically active metabolite by aldehyde ketone reductases. MHD is an active substance that plays a major pharmacological role in OXC activity [27,28,29,30]. In addition, MHD is metabolized through glucuronidation, a process mainly involving the enzymes UGT1A9 and UGT2B7 [32]; a small portion of MHD is also oxidized to the 10,11-dihydroxy derivative that is devoid of pharmacological activity [32]. Herein, we used the Michaelis–Menten kinetics of substrate consumption to characterize MHD metabolism. UGT1A9 accounts for 40–45% of the total metabolism, and UGT2B7 accounts for 15–20%. UGT1A4 contributes only a small and negligible fraction. The metabolic fractions of UGT1A9 and UGT2B7 were previously reported to be 0.45 and 0.2, respectively [32]. The Michaelis–Menten constant (K_m_) was obtained from the literature, while the k_cat_ value, a turnover number described as the maximum reaction rate (V_max_) per unit of the enzyme, was determined by fitting it to metabolic fractions.

### 2.3. Development of Pregnancy Population PBPK Models

In PK-Sim^®^ using clinical PK data search and TDM data collected from hospitals, the PBPK model of non-pregnant people was established and validated. Then it was transferred to MoBi^®^ to add pregnancy model structure and finally returned to PK-Sim^®^; the simulation was carried out in the pregnant population [17,25,26,27,28,29]. Changes in anatomical/physiological parameters associated with pregnancy have been fully described in previous studies [1]. At this stage, except for fraction unbound (*F_u_*) and enzyme clearance, the other parameters basically remained unchanged. *F_u_* (Equations (1) and (2)) in a specific week after fertilization was adjusted according to the initial value and the change in plasma albumin level with the duration of pregnancy [7,25,28].
(1)$$F_{u}=\frac{1}{1+K_{A}*P}$$
*P* = 14.7exp(−0.0454X) + 31.7
(2)

where *K_A_* is the equilibrium association constant, *P* is the drug concentration bound to protein (i.e., albumin), and X is the number of weeks after fertilization.

In women who are not pregnant, *K_A_* can be estimated using Equation (1) [16,17]. Assuming that *K_A_* and the number of binding sites are not affected by pregnancy, $F_{u}$ can be extended to women who are pregnant through Equation (2). Pregnancy-specific concentrations of albumin or α-1-acid glycoprotein (AAG) were used [16,17]. Neither OXC nor MHD is bound to AAG; however, MHD is bound to serum albumin.

### 2.4. Dose Simulations Based on the Pregnancy Model

The TDM range for MHD is 3–35 mg/L [33]. As a single-drug treatment, the maintenance-dose range of OXC is 600–2400 mg/d, and efficacy is realized in most patients on a daily dose of 900 mg/d [34,35,36,37]. We used the validated PBPK model for the simulation of dose optimization. According to the fertilization week range, three virtual pregnancy groups were created, namely the first three months of pregnancy (1–11 weeks), the second trimester (12–26 weeks), and the third trimester (27–38 weeks), with people who were not pregnant (20–40 years old) as the reference population. Each virtual population contained 1000 individuals [25].This model was used to predict the steady-state PK of OXC at different doses in women who were not pregnant and those who were pregnant, a scheme that was adjusted according to the dosage before pregnancy.

### 2.5. Parameterization of Placental Transfer

We used three different modalities to assess the two unknown parameters (D_pl_: passive fusion clearance through the placenta, K_f,m_: partition between fetal and maternal compartments) of placental transfer [18].

(1)In vitro cotyledon perfusion experiment

For estimates of transplacental transfer parameters, drug transfer across the placenta was modeled as a cotyledon divided into maternal and fetal compartments (Appendix A). Models of OXC and MHD transplacental transfer were investigated, namely simple diffusion, linear transfer, saturation transfer, and addition of placental elimination rate or tissue protein binding [36]. Briefly, we built a four-compartment model in MoBi^®^ with a slight modification of the ordinary equation system and used it to describe observational data,as reported in previous studies [38,39,40]. We assumed that there was no metabolism of OXC in the placenta. The time-dependent change in the molar drug amount in the four compartments, viz. maternal pool, maternal cotyledon, fetal cotyledon, and fetal pool (N{m, mp, fp, f}), was described using the following system of ordinary differential equations:

Maternal reservoir
(3)$$\frac{dC_{mp}}{dt}=\frac{Q_{m}}{V_{m}}*\left(\frac{C_{mp}}{KP_{pl}}-C_{m}\right)$$

Maternal cotyledon
(4)$$\frac{dC_{mp}}{dt}=\frac{(Q_{fm}*\left(C_{m}-\frac{C_{mp}}{KP_{pl}}\right)-\left(C_{mp}-C_{fp}\right)}{V_{mp}}$$

Fetal cotyledon
(5)$$\frac{dC_{fp}}{dt}=\frac{\left(Q_{f}*\left(C_{f}-\frac{C_{fp}}{KP_{pl}}\right)+D_{cot}\left(C_{mp}-C_{fp}\right)-K_{PE}*C_{fp}*V_{fp}\right)}{V_{fp}}$$

Fetal reservoir
(6)$$\frac{dC_{f}}{dt}=\frac{Q_{f}}{V_{f}}*\left(\frac{C_{fp}}{KP_{pl}}-C_{f}\right)$$
where *C* represents the concentration (mg/L), *Q* represents the flow rate (L/h), and *V* represents the volume (L). The subscripts *m*, *f*, and *p* denote the mother, fetus, and placenta, respectively. *KP_pl_* is the placental partition coefficient, *D_cot_* is the diffusion parameter (L/h), and *k_PE_* is the placental elimination parameter (h^−1^). *KP_pl_*, *D_cot_*, and *k_PE_* were estimated.

Estimation of in vitro “transdermal leaf transfer parameters” (*D_cot_*, *K_f,m_*) was achieved in the fetal–maternal PBPK model after scaling *D_cot_*to *D_pl_* using Equation (7) [18,38]:(7)$$D_{pl}=\frac{D_{cot}*V_{pl}}{V_{cot}}$$
where *D_pl_* (mL/min) and *D_cot_* (mL/min) represent transplacental clearance and transembryonic passive fusion clearance, respectively, and *V_pl_* (mL) and *V_cot_* (mL) represent placental and cotyledon volumes, respectively.

(2)Two other methods

Specifically, *D_pl_* was estimated according to the method proposed by Zhang et al. [41], where *D_pl_* was estimated from the permeability of the Caco2 cell line. Another approach was implemented by default in the Open Systems Pharmacology software, estimating permeability across organ membranes from physicochemical descriptors of lipophilicity and molecular weight.

### 2.6. Evaluation of the PBPK Models

The accuracy of the models was verified using published clinical data. The simulation conditions of the models were consistent with the research conditions reported in the clinical trials. The observed in vivo concentration below the lower limit of quantification (LLOQ) was considered “half LLOQ”. Model validation included the fold errors of the blood concentration–time curve and PK parameters (including AUC and C_max_). The fold error of each PK parameter was calculated as follows:(8)$$\mathrm{Fold}\;\mathrm{error}=\frac{\mathrm{Predicted}}{\mathrm{Obeserved}}$$
where Predicted represents the predicted model value and Observed represents the measured clinical value or observed value. If the fold error was between 0.5 and 2.0, the prediction of the model was considered acceptable [39,42]. Other visual prediction checks included goodness of fit plots in which individual in vivo concentration values at each time point were converted to geometric mean values [28].

## 3. Results

### 3.1. OXC Model Development and Verification

The measured data [19,20,21,22,23,24,43] of single-dose and multi-dose oral OXC in people who are healthy were fitted with the results predicted by the PBPK model. The dose scheme and demographic characteristics of these adults are shown in Table 2. As depicted in Figure 1, most of the measured data points of blood drug concentration are distributed on both sides of the blood drug concentration–time curve predicted by the PBPK model, and the predicted and actual values fit well. As shown in Table 3, the PK parameters predicted by the model, including AUC, C_max_, and T_max_, are basically consistent with the measured values. According to the collected clinical data, as shown in the goodness of fit diagram (Figure 1J), more than 95% of the predicted drug concentrations are within the error range of the measured values.

### 3.2. PK Prediction in the Pregnant Population

Steady-state PK simulations were performed during the first (6 weeks), second (20 weeks), and third (34 weeks) trimesters and compared to those of baseline levels [25]. The predicted values of the normalized concentrations of total and unbound OXC doses during pregnancy were close to the measured values, with errors ranging from 0.8 to 1.2 (Table 4). UGT2B7 and AKR1C enzyme activities remained unchanged [44,45], and UGT1A9 was upregulated [45], but the overall change was not statistically significant. The simulated dose scheme and demographic characteristics of the women who were pregnant are shown in Appendix A. The model was validated using clinically measured data [8,46]. The PBPK model accurately predicted the steady-state PK of maternal plasma OXC and MHD during pregnancy (Figure 2 and Appendix A).The average plasma concentration at the 300 mg/d dose fluctuated but basically remained stable; however, it was not guaranteed that the lowest effective therapeutic concentration could be reached at any time during the dosing interval in the third trimester (18.4 μmol/L). Throughout pregnancy, the steady-state C_min_, C_max_, AUC_τ-ss_, and t_1/2_ of OXC and MHD varied slightly (varied by no more than 50%) (Table 5). The model predicts that the average steady-state trough concentration of MHD decreases by 15.9%, 34.9%, and 41.5% in the first, second, and third trimesters (1st trimester, 2nd trimester, and 3rd trimester), respectively (Table 5). This predicted decline is slightly higher than the observed decline calculated from the TDM data (Figure 2d).

### 3.3. Dose Simulation in Pregnancy

The main PK parameter data for different doses during pregnancy are shown in Table 6. According to the simulation results, the monitoring frequency using TDM can be increased in the first trimester, and the drug concentration can be adjusted according to the baseline dose in the second and third trimesters. The PBPK model showed increased dose adjustment in the last two trimesters. We generally recommend that women take at least 1.5 times their baseline dose of OXC during the second and third trimesters. If efficacy is still not achieved at 2400 mg/d, continuing to increase the dose is not recommended because most patients cannot tolerate higher daily doses, and no systematic studies have been conducted on daily doses greater than 2400 mg. In those situations, it is necessary to consider changing the medicine or combining medications. As depicted in Figure 3 and Figure 4, a dose of 300 mg/d administered during pregnancy may be lower than the minimum effective dose and will lack efficacy. The suggested minimum dose during pregnancy is 600 mg/d.

### 3.4. Comparison of Different Estimation Methods of Placental Permeability

The results of different methods for evaluating the placental permeability of MHD are as follows: (a) in vitro cotyledon perfusion method [1], 129.62 mL/min (the ratio of cotyledon permeability to placental permeability according to Equation (2)); (b) the method suggested by Zhang et al. [48], 140 mL/min; (c) Open Systems Pharmacology software default method, 199.42 mL/min. Figure 5 compares the simulated concentrations in an in vitro human perfusion model. The parameter estimates are summarized in Table 7. In order to illustrate the effect of these differences on predicted fetal concentrations, the umbilical vein PK profile and observational data are presented in Figure 6. As shown in Table 8, the results of the in vitro cotyledon perfusion experiment are within the acceptable range, and the placental transport parameters obtained by this method seem to be more suitable for the molecules investigated herein. The results showed that fetal umbilical vein blood concentration during pregnancy was similar to that of the maternal vein during delivery, and the average umbilical vein concentration predicted by the ex vivo cotyledon perfusion approach was 20.64 μmol/L.

Oxcarbazepine was simulated in 100 pregnant women (300 mg, bid). The results showed that the mean steady-state concentration of MHD (C_ss,avg_) was 6.04 mg/L in non-pregnant women, while C_ss,avg_ were 4.92, 3.68, and 3.56 mg/L in 1st trimester, 2nd trimester, and 3rd trimester, respectively. The ratio of fetal umbilical venous blood concentration to maternal venous blood concentration (C_f_/C_m_) in 1st trimester is 0.89, and 0.87 and 0.94 in 2nd trimester and 3rd trimesters, respectively.

## 4. Discussion

PBPK models are increasingly being used in clinical settings for predicting PK in special populations. In this study, we (1) initially established a PBPK model for MHD in adults, (2) developed a PBPK model for women who are pregnant based on the dosing recommendations of OXC in those with epilepsy (Table 6), and (3) established different methods to predict placental transport permeability for comparative evaluation.

Based on previous PK studies, it is speculated that the increased intrinsic clearance of MHD throughout pregnancy may be due to increased glucuronidation of MHD, increased renal excretion, decreased absorption/intake of OXC, decreased binding of MHD to proteins, and increased volume of distribution [6,49]. OXC is almost completely absorbed and rapidly metabolized to MHD after oral administration, with little change during pregnancy. Changes in protein binding may have had an effect, but the magnitude of the change was not as high as expected due to the low protein binding of the drug itself [2]. The volume of distribution typically increases during pregnancy, but since the mean concentration of steady-state MHD is independent of the volume of distribution, this is not expected to affect the concentration of MHD [20]. Combined with the above analyses, increased glucuronidation and increased renal excretion of MHD are the most likely major contributing factors.

This study investigated the PK of OXC in women who are pregnant and the findings provide a reference for drug adjustment during pregnancy. To the best of our knowledge, this is the first pregnancy model of OXC. According to the model that was established, we postulate that one of the main reasons for the changes in OXC PK may be that the enzyme expression of UGT1A9 changes with pregnancy (Appendix A) [21]. However, further clinical trials are needed to confirm our speculations. While the overall magnitude of the simulated change is acceptable (<50%), steady-state plasma concentrations may drop below therapeutically effective concentrations (Figure 4).

We used our PBPK model to predict PK during pregnancy. The simulation results showed that with an increase in the gestational period, both the C_min_ and AUC of OXC decreased to varying degrees (Table 5). The PK was not significantly different between the non-pregnant and pregnant populations in the first trimester, but it was significantly affected in the second and third trimesters (Figure 2). This increases the risk of seizure frequency, as similarly reported in previous studies [7,9]. Since the therapeutic effect of OXC is related to its concentration, dose adjustments can also be made based on the preconception baseline dose (Table 5 summarizes the adjustment strategy).

We used three methods to predict the unknown parameters D_cot_ and K_f,m_ during placental transport and applied them to the PBPK model to simulate fetal umbilical venous blood concentrations at delivery (Figure 6). The results showed that the average umbilical vein concentration predicted by the in vitro method was 20.64 μmol/L, and the average concentration of the measured data was 18.94 μmol/L [49]. Experimental simulations of in vitro cotyledon perfusion may be more suitable for predicting the maternal–fetal PK of OXC than the other two methods. However, more complete study data are still needed for verification and analysis. Based on previous studies, fetal umbilical venous blood concentrations were similar to maternal venous concentrations at delivery, with a *C_f_*/*C_m_* ratio of 0.61–0.92 [49,50]. The ratio of simulated fetal umbilical venous blood concentration to maternal venous blood concentration (*C_f_*/*C_m_*) determined using the Caco-2, MoBi, and in vitro methods was 1.31, 1.16, and 0.62, respectively. Regardless of the method, all values above indicate that OXC and its active metabolites have good placental permeability, and the higher lipid solubility of MHD may explain why the concentration of OXC in fetal circulation reaches that in maternal circulation earlier than the concentration of MHD does. However, the placental transfer of OXC requires more in-depth research.

As shown in Figure 6, the results obtained by three different methods do not seem perfect. There are many reasons for this. First, it is reported that the human placenta may express a small amount of glucuronidase, which metabolizes OXC into MHD in the placenta [1,51]. However, for the lack of the amount data on glucuronidase in the placenta, the placental metabolic effect could not be included in the model at present. One study indicated that the metabolism of OXC to MHD in the placenta had only a trace level [51], which indicated that such an effect might be little. Second, the UGT1A9 gene polymorphism factor was not included in the model, also for the lack of expression data in the placenta and fetus of different UGT1A9 genotypes during pregnancy. Third, the transporter factor was not included in the model. Both oxcarbazepine and MHD are substrates of P-gp transporters [52], which indicate that P-gp might mediate placental exocytosis to a certain extent, although many studies indicated that oxcarbazepine and its active metabolite were mainly passively transported on the placenta due to their high-fat solubility [1,49].

Our study has some limitations, and further research is needed to improve the model. First, there were little clinical validation data for the pregnancy model, and our final model validation was insufficient. Second, the change in AKR1C1-4 enzyme activity remained unclear, which may have affected the simulation results. Third, OXC and MHD are both substrates of P-gp. The effect of P-gp was not included in our model. However, due to the high-fat solubility, the attribution of P-gp on the transportation of OXC and MHD in the placenta might be little. Fourth, the prediction of fetal exposure was not perfect, and more placental transport and metabolism data are needed to improve the model. The observation data of the umbilical vein during fetal delivery only used steady-state concentration, and complete concentration–time profile data for verification were unavailable. Despite these limitations, the model could predict maternal and fetal PK of OXC monotherapy and guide epilepsy patients regarding the use of OXC.

## 5. Conclusions

We successfully developed and validated a maternal–fetal physiological PK model for OXC. The dosing simulation results suggest that the OXC dose in the final two trimesters needs to be adjusted. We generally recommend that women take at least 1.5 times their baseline dose of OXC during the second and third trimesters. In addition, we successfully predicted that MHD exposure in maternal venous blood is similar to that in fetal venous umbilical cord blood. These models can enhance the understanding of the maternal–fetal PK behavior of drugs and may be useful tools for generating MHD PK predictions and supporting dose adjustment or other relevant decision-making scenarios in the clinical setting.

## Figures and Tables

**Figure 1 pharmaceutics-14-02367-f001:**
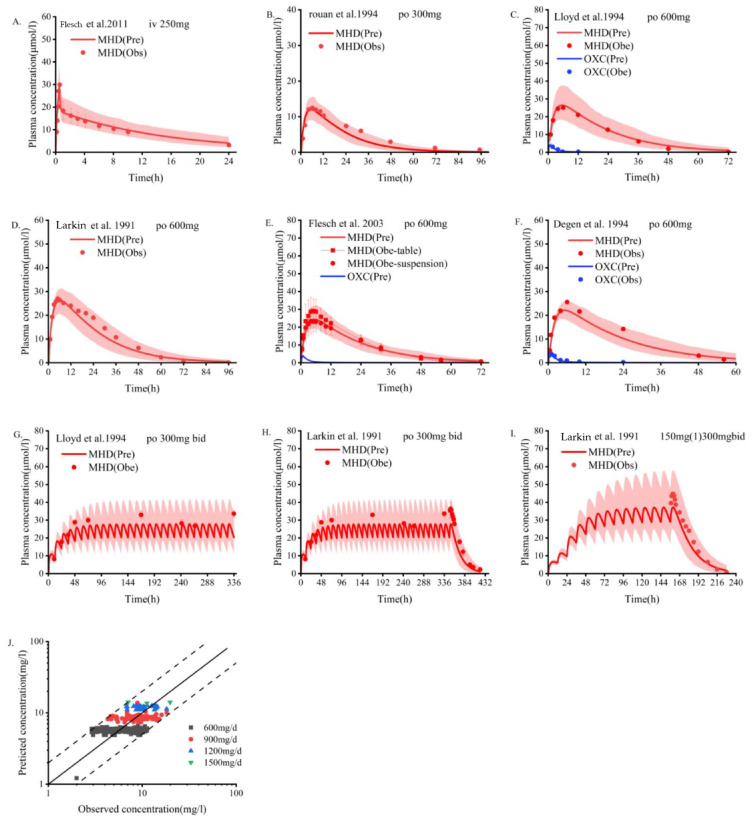
Population PBPK simulations (**A**–**I**) predicting median plasma concentrations of OXC in the non-pregnant population are depicted as dark lines, with shaded areas representing the 5th to 95th predicted range. Solid points are mean plasma concentrations extracted from clinical studies. Blue indicates oxcarbazepine; red indicates MHD. (**A**) Intravenous MHD; (**B**–**I)** Oral OXC; (**J**) Goodness-of-fit plots predicted by the OXC plasma concentration model. Different colors represent data from different doses. PBPK, physiologically based pharmacokinetic; OXC, oxcarbazepine; MHD, 10,11-dihydro-10-hydroxy-carbazepine [19,20,22,23,24,43].

**Figure 2 pharmaceutics-14-02367-f002:**
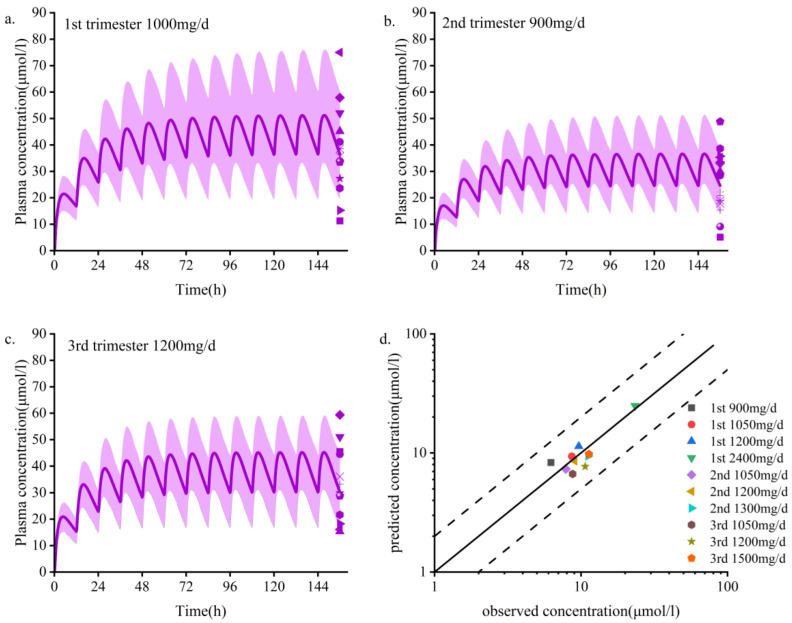
Population PBPK simulations (**a**–**d**) predicting median plasma concentrations of OXC in the pregnant population are shown as dark lines, with shaded areas representing the 5th to 95th predicted range. (**a**–**c**) First, second, and third trimester periods; d. Goodness-of-fit plots predicted using the MHD plasma concentration model for women who are pregnant. Different colors represent data from different doses. PBPK, physiologically based pharmacokinetic; MHD, 10,11-dihydro-10-hydroxy-carbazepine. Data sources: Yin et al. [8] and Christensen et al. [46].

**Figure 3 pharmaceutics-14-02367-f003:**
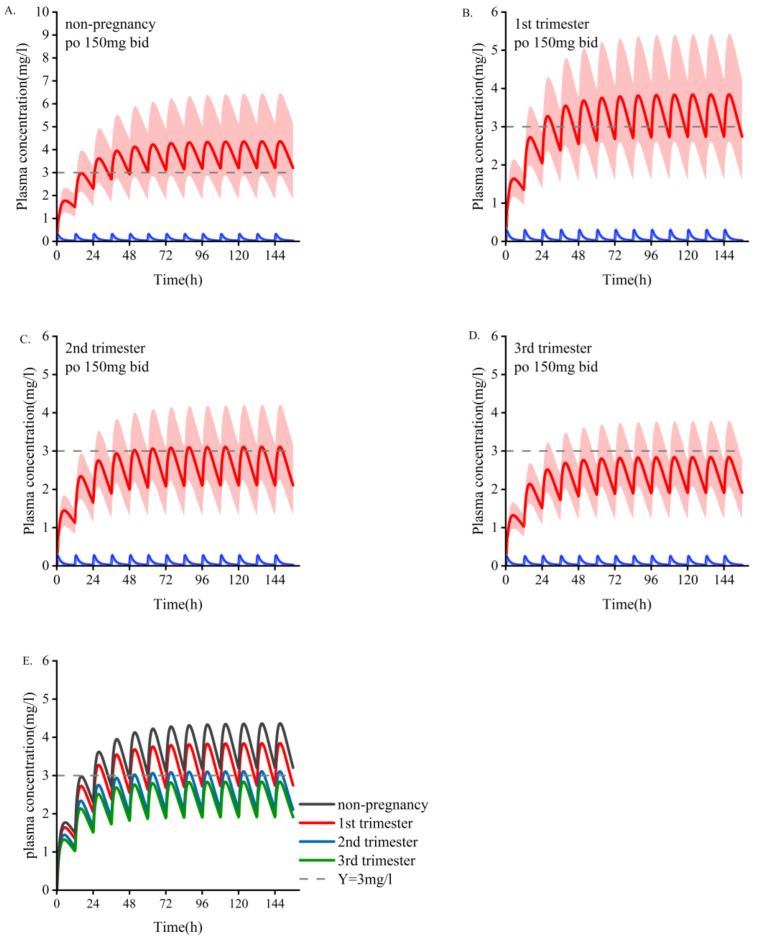
Simulate the pharmacokinetics of oral oxcarbazepine 150 mg twice daily in women who are pregnant and those who are nonpregnant. Blue represents oxcarbazepine; red indicates the metabolites of MHD. (**A**) The average age is 30 years old. (**B**–**D**) Women in the early, middle, and late stages of pregnancy. (**E**) Comparison of plasma concentrations of the active part of oxcarbazepine under the above four administration conditions. The mean concentration curve is represented by a dark line, and the shaded area represents the standard deviation. The dotted line indicates the lower limit of the recommended treatment range of oxcarbazepine (3 mg/L).

**Figure 4 pharmaceutics-14-02367-f004:**
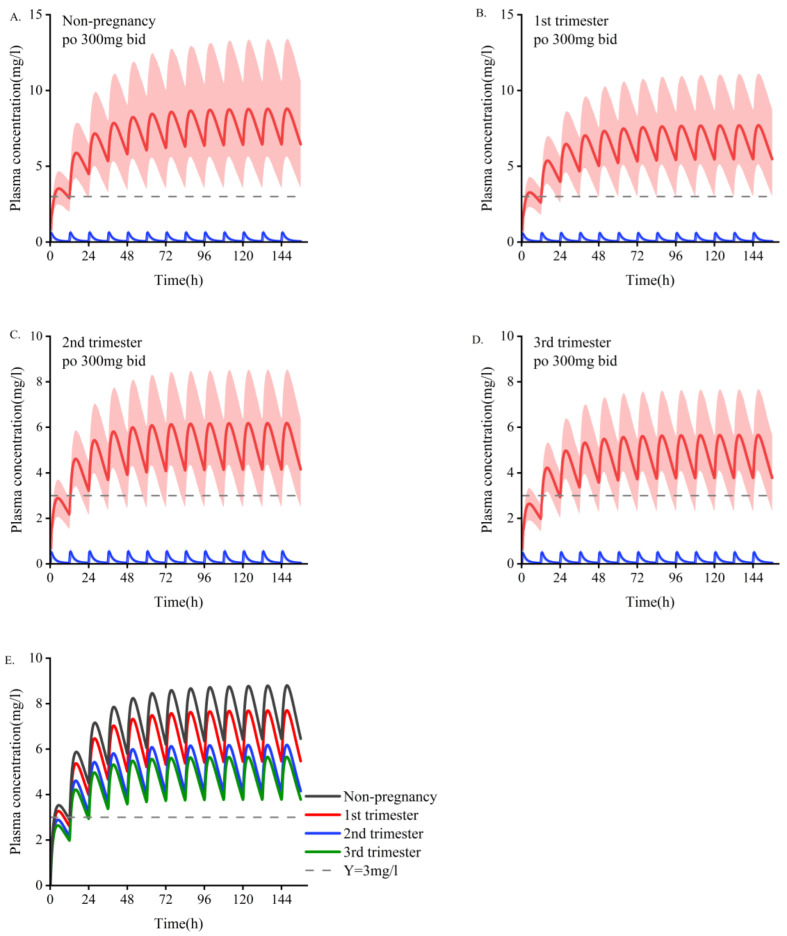
Simulate the pharmacokinetics of oral oxcarbazepine 300 mg twice daily in women who are pregnant and those who are nonpregnant. Blue represents oxcarbazepine; red indicates the metabolites of MHD. (**A**). Non-pregnant population with an average age of 30 years. (**B**–**D**). Women in the first, second, and third trimesters of pregnancy. (**E**). Comparison of plasma concentrations of the active fraction of oxcarbazepine under the above four administration conditions. The average concentration curve is displayed as dark lines, and the shaded areas indicate the standard deviation. The dotted line indicates the lower limit of the recommended treatment range of oxcarbazepine (3 mg/L).

**Figure 5 pharmaceutics-14-02367-f005:**
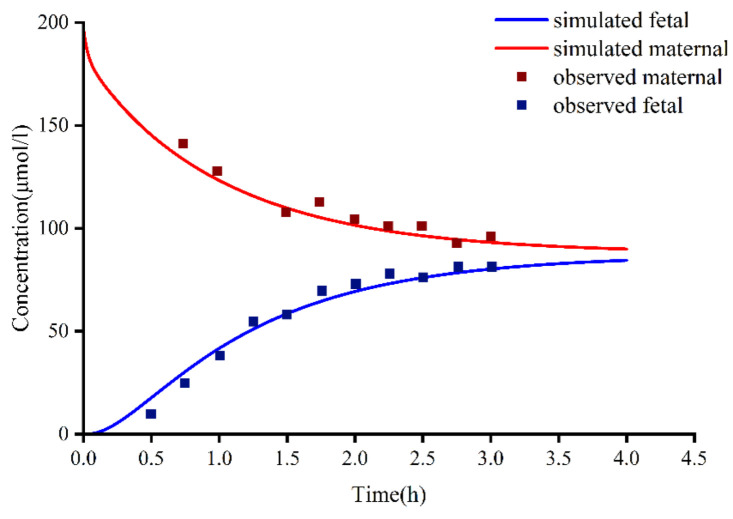
In vitro cotyledon perfusion experiments to observe fetal and maternal oxcarbazepine concentrations compared to fetal and maternal-simulated MHD concentrations. MHD, 10,11-dihydro-10-hydroxy-carbazepine.

**Figure 6 pharmaceutics-14-02367-f006:**
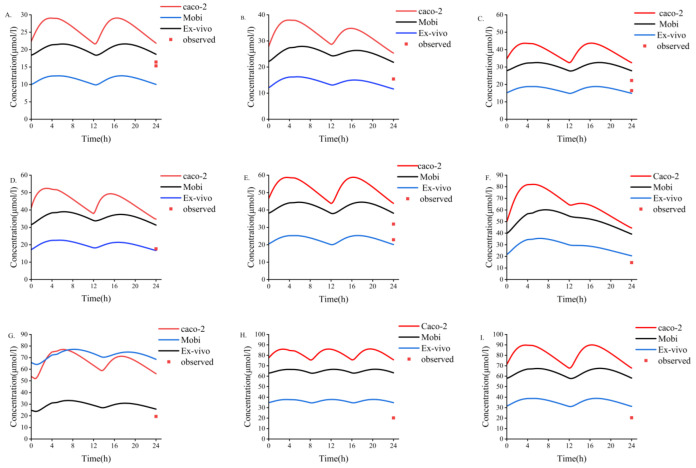
Three different methods used to predict fetal umbilical vein blood concentration during delivery. (**A**–**I**) Simulation of fetal umbilical vein blood concentration of MHD after oral administration of different doses of OXC. (**A**) 300 mg bid dose; (**B**) 300 mg qd +450 mg qd dose; (**C**) 450 mg bid dose; (**D**) 450 mg qd +600 mg qd dose; (**E**) 600 mg bid dose; (**F**) 300 mg qd +1200 mg qd dose; (**G**) 600 mg qd +900 mg qd dose; (**H**) 600 mg tid dose; (**I**) 900 mg bid dose. bid, twice daily; tid, thrice daily; qd, once daily. MHD, 10,11-dihydro-10-hydroxy-carbazepine; OXC, oxcarbazepine.

**Table 1 pharmaceutics-14-02367-t001:** List of reported clinical studies that were used for modeling adults who are healthy.

Reference	Drug	Form	Dose (mg)	Dosing Frequency	Weight (kg)	Age (Year)	Number of Participants (%Female)
Flesch et al., 2011 [19]	OXC	Tablets	300	Single	NR	NR	12 (50%)
	MHD	solution	250				
Lloyd et al. [20]	OXC	NR	600	Single	NR	NR	NR
			300 bid	Multiple			
pauline et al. [21]	OXC	Tablets	600	Single	47.6–80.7	60–79	12 (100%)
			150(1) 300 bid	Multiple			
Flesch et al., 2003 [22]	OXC	Tablets	600	Single	53–87	21–37	12 (0%)
		suspension					
Larkin et al. [23]	OXC	Tablets	300 bid	Multiple	70–89	22–43	8 (0%)
Degen et al. [24]	OXC	Tablets	600	Single	70–93	37–51	6 (0%)
Rouan et al. [24]	OXC	Tablets	300	Single	70 (mean)	24–35	6 (0%)

OXC, oxcarbazepine; MHD, mono-hydroxy derivative of OXC; NR, not reported; Concentrations of the S and R enantiomers of MHD were reported for both IV and oral doses. However, total MHD concentrations were used for modeling purposes, which were calculated as the summation of S-MHD and R-MHD concentrations. This was a reasonable approximation for modeling purposes since the pharmacokinetic properties of these two enantiomers are not significantly different, and also their anticonvulsant efficacy is similar. For 150 mg (1) 300 bid: first dose 150 mg, follow 300 mg bid.

**Table 2 pharmaceutics-14-02367-t002:** Summary of input data for the physiologically based pharmacokinetic models for oxcarbazepine and its metabolites, as implemented in the PK-Sim template.

Parameter (Unit)	Oxcarbazepine	References	10-Monohydroxy Derivative	References
Molecular weight (g/mol)	252.27	drug bank	254.28	drug bank
Lipophilicity (log units)	1.31	drug bank	1.2	drug bank
pKa (acid)	13.73	drug bank	NA	
Fraction unbound	0.35	drug bank	0.6	drug bank
Major binding protein	Unknown		albumin	
Solubility (pH 7) (mg/mL)	0.3	[15]	NA	
Weibull 50% dissolution time (min)	98.66	fitted	NA	
Intestinal permeability (transcellular) (cm/min)	4.45 × 10^−5^	[9]	3.79 × 10^−5^	fitted
Model for estimating organ-to-plasma partition coefficients	PK-sim Standard		Berezhkovskiy	
Metabolic clearance parameters				
CL_intAKR1C1_ (µL/min/pmol enzyme)	0.465	[9,17]	NA	
CL_intAKR1C2_ (µL/min/pmol enzyme)	0.53	[9,17]	NA	
CL_intAKR1C3_ (µL/min/pmol enzyme)	0.392	[9,17]	NA	
CL_intAKR1C4_ (µL/min/pmol enzyme)	0.13	[9,17]	NA	
V_max_ UGT1A9 (μmol/L/min)	NA		10.8	[32] ^a^
K_m_ UGT1A9 (μmol/L)	NA		150.07	
K_cat_ UGT1A9 (L/min)	NA		19.88	
V_max_ UGT2B7 (μmol/L/min)	NA		35.32	[32] ^a^
K_m_ UGT2B7 (μmol/L)	NA		3314	
Kcat UGT2B7 (L/min)	NA		5.66	
Renal clearance parameters				
CL_int,R_ (L/min)	0		0.05	[10,26] ^a^

CL_int,AKR1C1_ to CL_int,AKR1C4_, intrinsic clearance measured with recombinant aldo-keto reductase (family 1) isoforms C1–C4, respectively;UGT,uridine 5′-diphospho-glucuronosyltransferase;CL_int,R_, intrinsic clearance per unit kidney volume; ^a^, optimized value according to the original data in the literature.

**Table 3 pharmaceutics-14-02367-t003:** Observed and simulated pharmacokinetics parameters of OXC following oral administration of different dosing regimens in adults who were healthy.

Study	Dose	Sim	Methods	C_max_/C_max-ss_ (μg/mL)	AUC/AUC_τ-ss_ (μg·h/mL)	t_max_ (h)
Flesch et al., 2011 [19]	MHDiv250 mg	MHD	Predicted	7.36	58.51	0.50
			Observed	7.61	55.80	0.56
			FE	0.97	1.05	0.89
Rouan et al. [24]	OXC po300 mg	MHD	Predicted	3.07	80.06	5.25
			Observed	3.17	104.68	5.97
			FE	0.97	0.76	0.88
Lloyd et al. [20]	OXC po 600 mg	OXC	Predicted	1.08	3.34	0.65
			Observed	0.78	2.3	0.8
			FE	1.38	1.45	0.81
		MHD	Predicted	6.59	163.49	5.25
			Observed	6.37	161.56	5.96
			FE	1.03	1.01	0.88
Degen et al. [43]	OXC po600 mg	OXC	Predicted	0.89	3.07	0.75
			Observed	0.96	4.68	1.00
			FE	0.93	0.66	0.75
		MHD	Predicted	5.61	145.90	5.25
			Observed	6.50	175.84	6.06
			FE	0.86	0.83	0.87
pauline et al. [21]	OXC po600 mg	MHD	Predicted	6.67	199.11	5.00
			Observed	6.87	224.06	5.25
			FE	0.97	0.89	0.95
Flesch et al., 2003 [22]	OXC po 600 mg	MHD (Table)	Predicted	6.42	147.7	5.0
			Observed	7.4	173.59	5.0
			FE	0.87	0.85	1.0
		MHD (Sus)	Predicted	6.42	147.7	5.0
			Observed	5.93	165.05	4.0
			FE	1.08	0.89	1.25
pauline et al. [21]	OXC 150 mg (1) 300 mg bid	MHD	Predicted	9.47	267.23	3.50
			Observed	11.72	221.60	3.25
			FE	0.81	1.21	1.08

Sim: simulation; Sus: suspension; Observed: observed value; Predicted: predicted value; FE: Fold Error = Pred/Obs; 150 mg (1) 300 bid: first dose 150 mg, follow 300 mg bid.

**Table 4 pharmaceutics-14-02367-t004:** Total and unbound oxcarbazepine dose-normalized concentrations during pregnancy period.

Trimester	Total MHD			Unbound MHD		
	Observed	Predicted	Fold-Error	Observed	Predicted	Fold-Error
baseline	11.5	10.96	1.0	6.2	6.6	1.06
1st trimester	9.1	9.22	1.0	5.6	5.7	1.01
2nd trimester	7.8	7.13	0.9	4.3	4.6	1.07
3rd trimester	8.1	6.41	0.8	4.4	4.3	0.97

Dose-normalized concentration: steady state plasma concentration (mg/L)/100 mg dose.

**Table 5 pharmaceutics-14-02367-t005:** Mean steady-state trough concentration of oxcarbazepine predicted by the PBPK model and reported TDM data-based regression curve.

Method	Baseline Conc	6 Weeks Gestation		20 Weeks Gestation		34 Weeks Gestation	
		Conc	Change (%)	Conc	Change (%)	Conc	Change (%)
PBPK model	10.96	9.22	−15.9	7.13	−34.9	6.41	−41.5
Regression curve	11.5	9.1	−20.9	7.8	−32.2	8.1	−29.6

Conc: dose-normalized concentrations of MHD (μg/L/mg), data source: Page B. Pennel et al. [47].

**Table 6 pharmaceutics-14-02367-t006:** Changes in main pharmacokinetic parameters in women during different pregnancy periods compared with those of women of gestational age.

Dose Group	Trimester	AUC_0-inf_ (μg·h/mL)	AUCR	C_max,ss_ (mg/L)	CR (max)	C_min,ss_ (mg/L)	CR (min)
600 mg/d	Baseline	1092.23		8.80		6.46	
	1st trimester	975.09	0.89	7.7	0.88	5.47	0.85
	2nd trimester	766.74	0.70	6.18	0.70	4.15	0.64
	3rd trimester	700.33	0.72	5.66	0.64	3.78	0.59
900 mg/d	Baseline	1654.52		13.40		9.87	
	1st trimester	1444.02	0.87	11.67	0.87	8.3	0.84
	2nd trimester	1151.79	0.70	9.31	0.69	6.25	0.63
	3rd trimester	1050.99	0.64	8.51	0.64	5.69	0.58
1200 mg/d	Baseline	2238.48		18.22		13.52	
	1st trimester	1934.96	0.86	15.73	0.86	11.35	0.84
	2nd trimester	1563.68	0.70	12.66	0.69	8.56	0.63
	3rd trimester	1416.65	0.63	11.49	0.63	7.69	0.57
1500 mg/d	Baseline	2842.33		23.25		17.4	
	1st trimester	2468.07	0.87	20.09	0.86	14.49	0.83
	2nd trimester	1955.03	0.69	15.87	0.68	10.76	0.62
	3rd trimester	1777.89	0.63	14.45	0.62	9.74	0.56
1800 mg/d	Baseline	3458.85		28.44		21.48	
	1st trimester	2999.37	0.87	24.50	0.86	17.83	0.83
	2nd trimester	2368.88	0.68	19.27	0.68	13.17	0.61
	3rd trimester	2150.37	0.62	17.50	0.62	11.88	0.55
2100 mg/d	Baseline	4080.55		33.73		25.71	
	1st trimester	3536.62	0.87	29.00	0.86	21.30	0.83
	2nd trimester	2785.70	0.68	22.71	0.67	15.65	0.61
	3rd trimester	2523.45	0.62	20.57	0.61	14.08	0.55
2400 mg/d	Baseline	4699.85		39.04		30.04	
	1st trimester	4074.07	0.87	33.53	0.86	24.86	0.83
	2nd trimester	3201.03	0.68	26.14	0.67	18.18	0.61
	3rd trimester	2893.02	0.62	23.62	0.61	16.31	0.54

AUCR: the area under the drug duration curve of each pregnancy period (except for women of gestational age); CR(max): the highest plasma drug concentration in each pregnancy period was higher than that in pregnant women; CR(min): the lowest blood drug concentration in each pregnancy period was different from that in pregnant women.

**Table 7 pharmaceutics-14-02367-t007:** Values of placental transfer parameters calculated using three different methods, as implemented in a pharmacokinetic model based on fetal–maternal physiology.

Parameter	D_pl_ (mL/min)	K_f,m_
**OXC**		
Ex vivo cotyledon perfusion	597.27	0.75
Scaling of placental transfer rate via Caco-2cell permeability(according to Zhang et al. [41].)	1460	1.0
Scaling of placental transfer rate via physico- chemical properties(MoBi default method)	269.04	1.0
**MHD**		
Ex vivo cotyledon perfusion	129.62	0.40
Scaling of placental transfer rate via Caco-2 cell permeability(according to Zhang et al. [41].)	140	1.0
Scaling of placental transfer rate via physico- chemical properties(MoBi default method)	199.42	1.0

D_pl_ transplacental passive diffusion clearance; K_f,m_ partition coefficient between the fetal and maternal compartment.

**Table 8 pharmaceutics-14-02367-t008:** C_min_ ratio obtained after simulation during pregnancy at steady state (unit: μmol/L).

Dose (mg)	Scaling of Placental Transfer Rate via Caco-2 Cell Permeability			Scaling of Placental Transfer Rate via Physicochemical Properties			Ex Vivo Cotyledon Perfusion Experiment		
	Simulation	Observed	Ratio	Simulation	Observed	Ratio	Simulation	Observed	Ratio
300 × 2	21.86	15.89	1.38	18.66	15.89	1.17	10	15.89	0.63
300 + 450	25.35	15.42	1.64	21.82	15.42	1.42	11.6	15.42	0.75
450 × 2	32.5	19.35	1.68	27.84	19.35	1.44	14.9	19.35	0.77
450 + 600	34.76	17.7	1.96	31.33	17.7	1.77	16.88	17.7	0.95
600 × 2	43.81	27.39	1.60	38.15	27.39	1.39	20.11	27.39	0.73
300 + 1200	44.47	14.63	3.04	39.21	14.63	2.68	20.51	14.63	1.40
600 + 900	56.15	19.47	2.88	68.52	19.47	3.52	25.75	19.47	1.32
600 × 3	75.71	20.2	3.75	63.23	20.2	3.13	34.8	20.2	1.72
900 × 2	67.8	20.37	3.33	58.18	20.37	2.86	31.2	20.37	1.53

## Data Availability

The data that support the findings of this study are available from the corresponding author upon reasonable request.

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
