# Peer review of "Application of Physiologically Based Pharmacokinetic Modeling to Predict Maternal Pharmacokinetics and Fetal Exposure to Oxcarbazepine"

_pharmaceutics, 2022, doi:10.3390/pharmaceutics14112367_

Round 1

Reviewer 1 Report

Dear Editor

The authors developed and validated pregnancy related PBPK model for Oxycarbazepine (OXC). The dose of the OXC need to be adjusted as the gestation age progress. The major metabolites of the OXC, MHD concentration was similar in maternal venous blood as fetal venous umbilical cord blood. Authors suggested that this model can be used to predict the dose of MHD.

Overall comments

The manuscript should be edited by Scientific editing services. Many sentences were vague, and they were not scientific.

For example:

According to review and analysis of Arfman et al

to perform modelling work…This is just plain sentence. Please specific what exactly you have used scientifically.

Fully verified…what do you mean by fully verified. Please explain

Many places, the sentences needed to correct.

How different the metabolism of Oxcarbazepine to carbamazepine. Please explain

Mentioned potency of some AEDs…What are those AEDs...?

Significant placental transport may lead to excessive fetal plasma concentrations. What specific transporter proteins changed during pregnancy for OXC absorption?

Specific comments.

Have you considered Transporter proteins and metabolic enzymes changes while determined at different gestational ages? If you have considered it, please mention the values of those proteins.

Figures 3 and 4 mentions as women who are not pregnant and those who are…It would be better to mention nonpregnant and pregnant.

A validated PBPK model in non-pregnant is extrapolated to pregnant conditions to determine the drug concentration at fetus and maternal circulation. This must be emphasized and explained in better terms.

The model developed by the authors can be also used to determine the drug-drug interactions and how it was mentioned as limitation.

The rate of clearance from kidney and liver at different gestational ages must be mentioned. Change in levels of Transporter proteins and CYP enzymes have to be mention.

Reviewer 2 Report

I have read this paper with a background on clinical pharmacology commonly involved in PBPK work, but do not consider myself to be a hard core modeler. I have read this paper with great interest, I have suggestions to further improve the current paper.

First, in the introduction, there is a lot of focus on ‘anatomic’ teratology, but this topic is in my opinion somewhat broader ? Perhaps Veroniki et al, BMJ Open 2017 may assist you on this.

How were the clinical PK data for the model development and validation identified, divided and most importantly selected, as this can also result in biases (all steps of the chain, non-pregnant, pregnant and umbilical cord ? I feel uncomfortable (cf figure 1) at present in the absence of a clearly described search strategy: have all data been included, or were there ‘selected’ ?  

How well does this fit to the recent publication of a PBPK model for oxcarbazepine (Sinha et al, CPT Pharmacometrics Syst Pharmacol 2022, also a PK-Sim approach, ref 19 of the current paper)

The fold error (0.5-2.0) is quite broad, if you also consider the TDM target range previously suggested.

The blue ? lines in figure 3 are not yet described in the legend, perhaps the same holds true for figure 4 (likely fetal).

Looking at figure 6, it seems that the placental transfer models tested still performs rather poor. I assume that this has likely to do with the assumption. Assuming that a suggestion to perform in vitro placental transfer studies is too bold, could the authors at least explore the literature on likely mechanisms involved (active transport mechanisms, placental metabolism, or other mechanisms). Furthermore, can the authors also further elaborate on the fetal exposure (so before delivery ? as has been done for different drugs throughout pregnancy, including based on the PK-Sim approach, eg paracetamol, Mian et al, PMID 32052378).

Round 2

Reviewer 1 Report

Dear Authors

Thank you for addressing my comments. I can accept this version of the manuscript in the presented form. 

Reviewer 2 Report

no additional comments